# Advancements of Artificial Intelligence in Liver-Associated Diseases and Surgery

**DOI:** 10.3390/medicina58040459

**Published:** 2022-03-22

**Authors:** Anas Taha, Vincent Ochs, Leos N. Kayhan, Bassey Enodien, Daniel M. Frey, Lukas Krähenbühl, Stephanie Taha-Mehlitz

**Affiliations:** 1Department of Biomedical Engineering, Faculty of Medicine, University of Basel, 4123 Allschwil, Switzerland; 2Roche Innovation Center Basel, Department of Pharma Research & Early Development, 4070 Basel, Switzerland; vincent.ochs@unibas.ch; 3Department of Surgery, Canntonal Hospital Luzern, 6004 Luzern, Switzerland; leosnihat.kayhan@luks.ch; 4Department of Surgery, Wetzikon Hospital, 8620 Wetzikon, Switzerland; bassey.enodien@gzo.ch (B.E.); daniel.frey@gzo.ch (D.M.F.); 5Oncosurgery Zurich, 8802 Kilchberg, Switzerland; info@onkochirurgie.ch; 6Clarunis, University Centre for Gastrointestinal and Liver Diseases, St. Clara Hospital and University Hospital Basel, 4002 Basel, Switzerland; stephanie.taha@clarunis.ch

**Keywords:** artificial intelligence, liver diseases, surgeries, diagnosis

## Abstract

*Background and Objectives:* The advancement of artificial intelligence (AI) based technologies in medicine is progressing rapidly, but the majority of its real-world applications has not been implemented. The establishment of an accurate diagnosis with treatment has now transitioned into an artificial intelligence era, which has continued to provide an amplified understanding of liver cancer as a disease and helped to proceed better with the method of procurement. This article focuses on reviewing the AI in liver-associated diseases and surgical procedures, highlighting its development, use, and related counterparts. *Materials and Methods:* We searched for articles regarding AI in liver-related ailments and surgery, using the keywords (mentioned below) on PubMed, Google Scholar, Scopus, MEDLINE, and Cochrane Library. Choosing only the common studies suggested by these libraries, we segregated the matter based on disease. Finally, we compiled the essence of these articles under the various sub-headings. *Results:* After thorough review of articles, it was observed that there was a surge in the occurrence of liver-related surgeries, diagnoses, and treatments. Parallelly, advanced computer technologies governed by AI continue to prove their efficacy in the accurate screening, analysis, prediction, treatment, and recuperation of liver-related cases. *Conclusions:* The continual developments and high-order precision of AI is expanding its roots in all directions of applications. Despite being novel and lacking research, AI has shown its intrinsic worth for procedures in liver surgery while providing enhanced healing opportunities and personalized treatment for liver surgery patients.

## 1. Introduction

The world is now comprehending the recent developments in artificial intelligence, which originated in the 20th century. In 1956, an American computer scientist, John McCarthy, during his summer research project in 1956, created the term artificial intelligence (AI); however, this has its foundations as a concept dating back to the renowned Alan Turing. Mathematically, AI can thus be thought of as the union of numerical calculations performed with the assistance of a computer, with the intent of creating a form of intelligence. A faction of writers like to imagine that AI brings into being the simulations also performed on a computer, but with a triangulation in its objectives of analyzing, understanding, and predicting [1].

Most of us now use the terms artificial intelligence, machine learning, and deep learning interchangeably but, in fact, AI is a broad field that constitutes machine learning (ML) as a subfield that constitutes deep learning (DL), as shown in Figure 1. Machines can learn and comprehend from experience, the very same way in which humans do within an ML environment. ML relies on the tendency to perform tasks centered on algorithms and without actual programming. It can be further classified into unsupervised, supervised, and reinforcement learning [2]. DL is a kind of ML based on artificial neural networks in which various layers of processing are deployed to retrieve increasingly higher-level characteristics from data.

Even though many years have elapsed since AI originated, it has continued to undergo remarkable advances for its use in the medicinal field. It is now also used for therapeutic, diagnostic, and prognostic cases in almost every available field. Its use in the branch of medicine involving organs, such as the gallbladder, liver, pancreas, and the biliary tree, is especially useful. Hepatocellular carcinoma (HCC), a very common liver tumor, has been the main beneficiary in the area of hepatology, especially due to the fact that it reveals radiological characteristics, allowing for its diagnosis without the prerequisites for a histological study [3].

Over time, the advent of AI in the treatment of liver cancer has been made more comprehensible as a result of instituting the exactitude diagnosis and treatment systems, which have not only furthered the understanding of liver cancer amongst people, but have also qualitatively improved the diagnosis and treatment techniques of liver cancer. Novel computer technology as represented by AI has been utilized in the forecasting, screening, diagnosis, and treatment of liver cancer. The continual advancement of AI has provided liver surgery a new lifesaving tool, while also providing individualized treatment with enhanced experiences and greater healing chances for patients. Various researchers have proposed that most current applications and development in AI can be examined from six main aspects. These are virtual assistants, adjuvant therapy, medical imaging diagnosis, risk and treatment response prediction, post-operative rehabilitation management, and drug treatment and testing. The most gratifying developments and achievements of AI are argued to be medical imaging diagnosis and adjuvant therapy [4].

## 2. Pattern Recognition from Data

In pre-operative liver surgery, supervised learning allows the machine learning algorithm to achieve a pre-designed result, as opposed to unsupervised learning, which does not have any pre-defined output categories. Unsupervised machine learning, on the other hand, may find minute patterns in vast datasets during inspection methods that would be indiscernible to a trained eye. The use of supervised machine learning during disease establishment in liver pre-operative surgery allows datasets to be divided into training sets, from which tests are performed to assess the algorithm’s performance using the fresh dataset. Conditioning and reinforcement are closely related since they both entail repetitive trial and error scenarios that finally result in a reward. When it comes to deep learning (DL), neural networks and their combinations are used to learn complicated patterns, allowing for the learning of increasingly complex patterns [5].

To attain the best relationship between the output and input layers within AI, there exists multiple intermediate and hidden features, which are fed with data from previous layers, and which therefore influence any possible outcome. There have been instances in which the deep neural networks (DNN) utilizes electronic health records (EHR) and still manages to accurately predict mortality or anastomotic leaks within post-operative patients [6,7]. A DNN is a collection of neurons organized in a sequence of multiple layers, where neurons receive as input the neuron activations from the previous layer, and perform a simple computation (e.g., a weighted sum of the input followed by a nonlinear activation) in order to process data. Automated artificial intelligence models fed by livestreaming EHR data with mobile devices would need data uniformity, improved model interpretability, careful deployment and monitoring, consideration of algorithm bias and error accountability, and preservation of bedside assessment and human intuition. In deep learning, the more commonly used architecture includes conventional neural networks, residual networks, and recurrent neural networks. Herein, the techniques that AI use include natural language processing (NLP) and computer vision (CV). CV permits AI to identify every possible object that may be deemed necessary during surgery via processing previously available images and developed patterns. On the other hand, NLP permits machines to examine human language past their code or machine language. Hence, NLP and CV contribute to diverse applications in anesthesia and surgeries [8].

## 3. Development in Modern Surgical Methods

Modern surgery practices hedge their success on incessantly updating and renewing approaches and techniques via the use of exhaustive studies, peer reviews, and trials. Influenced by cutting-edge technologies and inspired by the potential of widespread future trends and the uncertainties of patient outcomes, it is necessary that current practices should be improved further. In analyzing any potential purview of surgery, it would be prudent to also broadly define what this “future” is, bearing in mind the probability of any ideal conditions occurring, simultaneously. The landscape that defines modern surgery would definitely be different in a decade’s time. Much like any technological advancements, it is important that there exists the initial scientific breakthrough with an agenda for paving the path for its development. To have space travel, jet propulsion was needed. To have the internet, there was a need for electricity. It is this comparative factor that also shows that the same is also the case if medical breakthroughs are to be attained. In realistically assessing the future of surgery, it is crucial that there is a cognizance that challenges still exist and which must be overcome [9].

A radiomics-based approach to extracting massive datasets from them cannot be disregarded by AI researchers [10]. Details of the same are discussed under Section 7.

A similar study performed by Holmgren G, et al. included all new adult intensive care admissions in Sweden from 2009–2017. Cross-validations were used to choose the best artificial neural networks (ANN—similar to DNN but they differ in the number of layers and the way each combination is constructed in order to identify patterns in datasets) from two hidden layers using random hyper-parameters. Similar to the Simplified Acute Physiology Score (SAPS)-III model, ANNs were created. The SAPS was developed to provide a measure of a patient’s physiological condition in clinical trials. In this way, groups of patients can be compared in clinical studies with regard to their general health status, e.g., in order to be able to detect the influence of different disease severity in different study groups. The parameters such as area under the receiver operating characteristic curve (AUC) and Brier scores were devised and used to evaluate the performance. The AUC is a probability curve that plots the true positive values against the false positive values at various thresholds. It therefore is used to separate the signal from the noise in a dataset. The constructed eight-parameter ANN model performed as well as SAPS-III but with superior calibration. The ANN model also better corrected mortality for age. For 217,289 intensive care unit patients, ANNs have been shown to outperform the SAPS-III model for 30-day death prediction [11]. AI-based decision analysis and reinforcement learning can help add higher value to surgeries by enhancing judgment and decision making.

## 4. Liver Surgery

Despite the technological advancements in image processing and development in the previous few decades, two-dimensional images are not sufficient to define the anatomical complexities of the human organ and their internal structure. Any reflection of abnormality or its successful identification from these generated images is beyond imagination. The advancement of liver surgery is founded upon the functional study of the hepatic vascular system, which comprises the hepatic vein, hepatic artery, biliary system, and portal vein. The breakthrough point of such surgeries involves understanding the complexities and variabilities of the liver’s internal structure.

Medical intelligence findings and treatment technologies have put tremendous stimulus on the advancement of liver surgery, propelling the procedure into the epoch of “intelligence”. It is therefore imperative for surgeons to share the available innate knowledge of medical intelligence and treatment technology advancements in medicines with data scientists. Knowing how to acclimatize AI for clinical practice is the best path surgeons can follow. While various aspects of digital medicine and intelligent diagnosis combined with treatment technologies are limited, they still provide information for learning that may have an impact on surgeons in the future. A brief correlation between surgical terminologies and DL is shown in Figure 2. Such aspects include, but may not be limited to, 3D virtual and visualization simulation surgery, virtual reality, 3D printing, molecular fluorescence imaging technology, navigation technology associated with abdominal surgery, artificial intelligence-imaging omics, and fresh tumor imaging technology–photoacoustic imaging [12].

## 5. Surgical Decision-Making Challenges

Time restrictions and ambiguity about diagnosis and treatment response sometimes affect surgical decision making. Conventional methods adopted to make a decision while in surgery always involve time constraints and even some unforeseen complexities, as a result of which predicted responses are often impaired. Reinforcement learning plays an invaluable role in reducing such anthropogenic errors and provides optimum care [13]. Decision making during surgeries, such as for liver cancer, calls for ultimate care and is influenced by patient emotions, values, complexity, decision-making volume, time constraints, individual judgment, hypothetical deductive reasoning, and the nature of patient–surgeon interactions. With each of these factors, both ineffective and effective methods exist for dealing with them, and which ultimately lead to both negative and positive results, respectively.

Another surgical decision-making challenge is complexity. Surgical decision making involves a hypothetical–deductive model of decision making that governs the assessment of initial patient presentations in the development of a list of viable diagnoses. These diagnoses will usually be differentiated via diagnostic testing using AI or via a rejoinder to empirical therapy. Values and emotions are other challenges, which with undesired tests and treatments, both of which are extensions of which AI makes great use. Shared decision-making procedures that make use of AI provide for better patient compliance and satisfaction [14].

## 6. Use of Artificial Intelligence in Liver Surgeries

Surgery is the right treatment for liver tumors. Surgery to remove liver tumors is difficult due to intricate architecture and worries about functioning liver remnants. An understanding of liver anatomy is essential for a successful hepatic resection. Changing the surgical plan might have a significant influence on the result. The anatomy is so intricate that mental reconstruction from CT or MRI scans alone is challenging. Intraoperative display of preoperative imaging data in hepatic surgery has been a hot issue for decades. AI focuses on imaging and navigation to help with pre-operative planning and intra-operative assistance [15].

Empirical evidence revealed that varied methods of AI have been put into practice in various studies containing heterogeneous patient sample sizes. A good number of the said studies made use of AI–ML based algorithms to analyze the prognosticators of surgical intricacies. To realize the 21st century digital revolution, particular national, international, and bioethical standards are required, while gathering data. Effective AI requires avoiding false positives [16].

In another similar study conducted by Abbod et al., investigation on studies of artificial intelligence usage in urological cancer included studies on machine learning’s features and execution. Other methodologies, such as expert systems and neuro-fuzzy modeling systems are being explored by certain researchers in this subject. The use of artificial intelligence appears to be more accurate and exploratory than standard regression statistics for large datasets but there are certain properties of each AI approach. They find that, neuro-fuzzy modeling methods can overcome artificial neural networks’ lack of transparency [17].

In consideration for the fact that there exists the need to have more accurate predictions, experts have made comparisons of AI techniques with those of traditional linear models with the aim of optimizing treatment decision making. In spite of the fact that several prediction models have put into use post-operative as well as pre-operative variables, the models utilized have not proven to be accurate and safe for clinical decision making due to the fact that they require data that are only available post-resection or provided for other treatments. On the contrary, models known to only have pre-operative variables have been proven to be able to assist in treatment strategies within a preoperative setting [18]

The outcomes of liver surgery have drastically improved over the past decade with post-operative mortality declining to less than 5% from 20%. This improvement is largely associated with advanced pre-operative imaging related to AI as well as better peri-operative care and AI-associated surgical techniques. These AI-associated and aided procedures also provide for better anatomical comprehension of the anatomy and technological progression of intra-operative composition, management of complications, and early identification. In spite of all this, liver surgery procedures remain quite technical and complex, necessitating cautious pre-operative preparation, intra-operative implementation from skilled surgeons, nursing staff, anesthetists, and extended operative hours. With post-operative morbidity above 20%, and mortality rates of some complicated resections around 10%, there is a need to continue exploring and assimilating progressive and innovative technological methods such as AI into clinical practice if the outcome of this morbidity is to improve [19].

Post-operative care has an immense role in the success. Predicting the possible future reaction in the recipient’s body during and after the surgery is beyond contemplation for humans. Well-trained robots, based on critical algorithms, have changed this whole perspective. The waiting time for transplantation can be very precisely predicted by AI, thereby reducing the risks involved to a greater limit.

Artificial intelligence (AI) has been examined for its role in prognostic population risk stratification and clinical resolution backing systems, endorsing its use in the fresh era of ordinal medicine and defined surgery. A vast majority of AI uses are usually based upon machine learning, a technique that automatically learns and recognizes specific patterns in addition to coming up with useful decisions, which it deduces from available data. The component of deep learning is almost within the same technique, replicating the neural network and peripherals of the brain for data analysis. It is intriguing that the most prevalent representative of deep learning is the fact that it is hedged upon and utilizes real data, with the decision-making process performed with minimal human interposition. It is therefore the integration of such processes, merging into the various components for the delivery of liver surgery, that permit the advancement of both post-operative and oncological outcomes [20,21]. Thus, major types of ML and AI interventions together can help to innovate novel techniques to be used in surgeries. The machine learning types remain the same for other interventions such as liver disease diagnosis, prognosis, measurement, or prediction, as shown in Figure 3.

Robot-assisted surgery may be used to research and further develop the use of AI in surgical practice by providing complete telemetry and a sophisticated viewing console. Machine learning improves surgical skill development, surgical process efficiency, surgical guiding, and postoperative outcome prediction. Tension sensors on robotic arms and augmented reality approaches can assist in improving surgical outcomes and tracking organ movement. The application of AI in robotic surgery is predicted to improve future surgical training as well as the operative experience. Both strive to achieve precision surgery and hence improve surgical treatment. Using AI in master-slave robotic surgery may allow for a more gradual transition to autonomous robotic surgery [22].

Medical robotics have further revolutionized the surgeries, targeted therapies, rehabilitation, and hospital automation, in particular, developing robots for the least invasive procedures. The combination of imaging, sensing, and robotics might affect patient care pathways toward precision intervention and patient-specific therapy. Future robotic therapies are developing trends to make them more user-friendly, lighter, and more ergonomically-sound, as well as safer and more accessible to clinicians [23].

Recent steep advances in medicine continue to provide AI with much-needed relevance due to the role it plays in the support of clinical decision-making processes. As a result, AI is also increasingly being used for genomics, precision medicine, drug discovery, risk stratification, and imaging and diagnosis. Introduced more recently into surgical procedures such as liver surgery, AI possesses strong roots in navigation and imaging, with early techniques largely concentrating on computer-assisted interventions and feature detection for both intra-operative guidance and pre-operative planning. This is to say that AI has continued to play a crucial role in liver surgery technological advancement such as navigation, imaging, and robotic intervention [24]

## 7. Artificial Intelligence in Liver-Associated Diseases

### 7.1. Hepatocellular Carcinoma (HCC)

Hepatocellular carcinoma (HCC) is by far the most common form of liver cancer which affects approximately one in every 1,000 people. Histologically confirmed hepatocellular carcinoma is more common in persons who have chronic liver disease, such as cirrhosis.

The application of AI evidently continues to appear as a valid assistant to traditional statistics as a result of its ability to process large amounts of data, while discovering concealed interconnections between variables. In other research focused on characterization of lesions, classifying, and segmenting, several approaches were utilized to test the convolutional neural network (CNN) model accuracy. Basically, the structure of a classical convolutional neural network consists of one or more convolutional layers, followed by a pooling layer. In principle, this unit can repeat itself as often as desired; if there are enough repetitions, we then speak of deep convolutional neural networks, which fall into the area of deep learning. The studies showed (the value of CNNs in image analysis and early identification of HCC and liver masses) accuracy of the CNNs employed to segment and classify pictures of liver cancer to be excellent. Deep learning and AI continue to play a central role in a myriad of topics concerning liver cancer research. The identification and diagnostic discernment of malignant versus benign liver masses has been variously reviewed to propose the notion that AI had the ability to differentiate liver cancer, and especially HCC from other lacerations. This identification is better compared to other methods, such as radiology image inspections and Bayesian models [25].

As an alternative to surgery, radiomics uses electromagnetic radiations to treat the patients. The use of radiomics in AI cannot be ignored due to the methodology it uses in the extraction of large datasets of features from a myriad of medical images and utilizing data categorization algorithms. Depending on the styles used to calculate them from an AI perspective, categories can be transform-based and statistics-based, as Table 1 illustrates. Radiomics research related to liver disease is primarily concentrated on hepatocellular carcinoma (HCC), as illustrated in Table 2, in determining the various AI methodologies used in screening objectives.

### 7.2. Cholangiocarcinoma (CCA)

Cholangiocarcinoma (CCA) is the second most prevalent primary hepatic malignancy after hepatocellular carcinoma. The prognosis for these tumors is dismal regardless of type, location, or cause. It is only possible to cure CCA by total surgical resection, but some patients still have local recurrence or distant metastases. Now with the help of AI tools, we know that CCA has a molecular biology that has changed substantially over the years, with unique diagnostic and treatment methods [26].

For the purpose of identifying CCA, Wang et al. used the main detecting approaches of the classical ML and DL-based algorithms in their study. The most widely used detection approach is semi-automatic segmentation with support vector machine classifiers. DL is becoming increasingly popular in computer auxiliary diagnosis (CAD) systems due to end-to-end training. Insufficient medical training data limits deep learning’s accuracy. Improving clinical diagnosis and treatment of CCA by analyzing artificial intelligence approaches used in CCA is one of the main aspects of AI [27]

### 7.3. Intrahepatic Cholangiocarcinoma (ICC)

This rare primary liver cancer arises from intrahepatic bile duct epithelial cells, intrahepatic cholangiocarcinoma (ICC) is the rarest kind of cholangiocarcinoma, compared to cancers in the upper biliary tract or the common hepatic duct bifurcation. Due to the difficulty in detecting and treating ICC, most patients are discovered at an advanced or fatal stage [28].

In one study, the data of 417 ICC patients between 1997 and 2018 was considered; 293 of these matched the requirements. An artificial neural network using known risk variables as input nodes (ANN) was then compared to the most extensively used traditional grading method, the Fudan score. The ANNs with a large number of known risk variables beat risk scores with a small number of parameters. When models trained on large multicenter datasets are made publicly available, they may enhance therapy stratification [29].

An ML-based model was developed by Tsilimigras et al., to assess the pre-operative proportion of individuals who would benefit most or least after intrahepatic cholangiocarcinoma (ICC) resection. The model found discrete prognostic categories for individuals with varying outcomes. The greatest predictive markers linked with survival were the size of the tumors, albumin–bilirubin (ALBI) grade, and pre-operative lymph node status among 1146 patients having ICC resection. Machine learning developed four separate patient groups based on the above-mentioned predictive markers. Results proved that the pre-operative patient selection and risk stratification have great results on the application of survival decision trees [30].

### 7.4. Colorectal Cancer Liver Metastases (CRLM)

Early detection of colorectal cancer liver metastases (CRLM) risk subgroups using radiomics baseline CT scan analysis with excellent accuracy and in less than 5 min might provide patients with the highest opportunity for an early diagnosis, more effective therapy, and hence the best outcome. Radiomics has also shown tremendous promise in supporting radiologists in detecting CRLM from CT and MRI images, as well as defining tiny nodules of unknown nature. A faster diagnosis method might save time and money for individuals and healthcare organizations alike. Recent studies show that AI can also predict chemotherapy response, early local tumor development following ablation treatment, and patient health post-surgery or chemotherapy [31].

### 7.5. Fibrosis/Cirrhosis

Chronic liver disorders and cirrhosis cause 1.1 million deaths globally each year. In 2017, cirrhosis prevalence rose to over 122 million from 71 million in 1990. In addition to viruses such as hepatitis B and C, other causes of chronic liver disease, such as alcoholic liver disease, can also lead to cirrhosis [32].

A deep learning algorithm based on CT scans can grade liver fibrosis. A clinical retrospective analysis comprised of 496 CT images of individuals for liver assessments had histological data on liver fibrosis stage. The 396 images were used to train a deep CNN (D-CNN), while rest 100 images were utilized to test the trained network against the histological fibrosis stage. The deep learning CT image score for fibrosis (F_DLCT_) correlated with various stages of liver fibrosis (Spearman’s correlation coefficient (r) = 0.48, *p* < 0.001). In terms of F_DLCT_ scores, the areas under the characteristic curves were 0.74 (0.64–0.85), 0.76 (0.66–0.85), and 0.73 (0.62–0.84) for significant fibrosis, advanced fibrosis, and cirrhosis, respectively, with confidence intervals of 95% [33]. In a similar study, again using CT scans, a comparable study concluded that the deep learning systems (DLS) provides appropriate staging of liver fibrosis. The DLS for CT-based liver fibrosis staging was established utilizing 7461 CT scans from individuals with pathologically proven liver fibrosis. The DLS’s diagnostic performance was tested on 891 patients in independent test sets. Logistic regression was used to assess the impact of patient variables and CT method on DLS staging accuracy. The areas under the characteristic curves were 0.96, 0.97, and 0.95, and were found in the DLS test datasets for identifying substantial fibrosis (SF), advanced fibrosis, and cirrhosis, respectively. The DLS performed outstandingly compared to the radiologists’ interpretations [34].

The DCNN outperformed radiologists in diagnosing cirrhosis and calculating the METAVIR score using ultrasound pictures. A DCNN was trained using 13,608 ultrasonography pictures from 3446 patients who underwent surgical resection, biopsy, or transient elastography. As a benchmark, pathological specimens or transient elastography-derived METAVIR scores were employed. Initiating a four-class model, the algorithm’s diagnostic performance was tested on 266 internal patients with 300 pictures and 572 exterior patients with 1232 photos. The four-class model’s accuracy was 83.5% for internal and 76.4% for exterior test sets. The DCNN’s AUC = 0.857 was greater than the five radiologists’ AUC, ranged 0.656–0.816 (*p* < 0.05), utilizing the external test set [35].

### 7.6. Non-Alcoholic Fatty Liver Disease (NAFLD)

Liver biopsy is the standard method for diagnosing liver fibrosis and NAFLD. Noninvasive options for liver biopsy, such as ultrasonography, elastography, and clinical prediction scores, have had mixed results. To increase the efficacy of noninvasive diagnostic instruments, AI algorithms are being developed. However, before using AI-assisted technologies in the detection of chronic liver diseases, their performance must be validated [32].

In a study conducted by Pournik et al., demographic characteristics, diabetic history, INR, albumin, total blood count, AST, ALT, and other basic laboratory tests, exams and medical history were acquired from 392 NAFLD patients. The authors chose relevant variables using a feature extraction approach (Knime software) and MATLAB to create a neural network. The study’s findings suggest that neural network modeling may be a simple, noninvasive, and accurate way for diagnosing. The model’s sensitivity and specificity were 86.6 and 92.7 percent, respectively [36].

Several ML-based characterization methods have been implemented for fatty liver disease (FLD) detection. These ML systems use a large number of ultrasonic grayscale features but due to the disparity among grayscale characteristics and classifier types, these are computationally demanding and slow. Kuppili et al., worked on an ELM-based tissue characterization method for risk classification of ultrasound liver imaging. ELM trains a single layered neural network, which is only fed forward, making it quicker than traditional ML approaches. The ELM accuracy was 96.75% vs. 89.01% for ML, and the AUC was 0.97 and 0.91, respectively. Further tests revealed a mean reliability of 98% for ELM classifiers and a 40% speed advantage over ML. Authors tested the symtosis system utilizing two-class biometric face public data with 100% accuracy [37].

### 7.7. Liver Transplantation

Healthcare and liver transplantation are increasingly using machine learning (ML). Transplanting organs from a deceased donor can add intricacy in this discipline. Organ transplantation is not the only factor contributing to the success of surgery. Any component of organ transplantation has input and output variables, such as processing the available image, predicting key outcomes, diagnosis details, therapeutic algorithms, and precision, involved in therapies.

Artificial intelligence classifiers differ in how they build associations between input variables, choose data groups to train patterns, and forecast the output variables’ alternatives. This requires hundreds of classifiers. There are a number of clinical fields which can make use of the applications of artificial intelligence in organ transplantation. Transplant professionals need to increasingly employ deep learning-based algorithms to assist their recommendations in the future [38].

ML has several uses in pre- and post-transplant situations, such as predicting patient survival, graft rejection and failure, and post-operative morbidity risk. Despite limitations such as site-specific training needs, the need for further multi-center research, and clinical interpretability optimization barriers, ML’s great potential to improve patient care warrants further investigation [39].

Artificial intelligence can anticipate the outcomes of output variables by identifying correlations between input factors. Artificial neural networks, decision tree classifiers, random forest, and Bayes classification models are the most extensively used classifiers for organ transplants. Artificial intelligence is currently being explored in the fields of organ transplantation (e.g., the liver), very specifically in survival analysis, and oncology [40].

ANNs are impactful possibilities for identifying patterns that are too complicated for a doctor to understand, and they can make very close predictions on data, reaching 95% for 3-month’s graft survival. However, it is expected that yet to be implemented neural networks have the potential to overcome some of the limitations of ANNs, particularly the lack of knowledge about the variables they give. For example, random forest algorithms may boost confidence in using marginal organs and improve transplant outcomes [41].

With advanced age and significant comorbidities, several problems throughout the transplantation decision-making process need to be adequately addressed. Conventional approaches fail to leverage enormous datasets with hidden, non-linear correlations between demographics, clinical, laboratory, genetic, and imaging characteristics.

## 8. Invasive Surgery Minimization

Recently, conventional methodologies for surgeries, abnormality detection, medicines, drug development, dose allocation, all of which require many human resources, have been less reliable for their effectiveness. DL, on the other hand, can learn typical algorithms and predict results that can give better results in all domains of medicine. Finding the least risky way from a wide range of datasets is a key feature of this technology. Detecting any abnormalities by mere image processing or minute detections of muscle fiber movement can give a broader, yet better understanding, of the target organ.

For processes involving medical imaging of the liver, medical practitioners aim to identify, characterize, and observe diseases through the visual assessment of liver medical images. Sometimes, these visual assessments, many of which are based on experience and expertise, may also be inaccurate and personal. In this instance, AI has the propensity to make quantitative assessments via automatic recognition of the imaging information, rather than by qualitative reasoning [42].

As a result, AI is able to aid physicians in making more productive and accurate imaging diagnoses, which can greatly minimize the physicians’ workload. In this respect, the two main methods of AI widely used in medical imaging are currently the traditional machine learning algorithm and deep learning algorithms. Traditional machine learning algorithms mainly depend on the pre-defined concocted structures that describe in detail the inherent regular patterns in the extracted data and from the regions of interest containing explicit parameters according to expert knowledge. Task-related or meaningful features are then defined to be aligned with mathematical equations to be able to be quantified by AI-related computer programs [43].

On the other hand, deep learning algorithms, a subset of machine learning, are largely based on neural network structures that are enthused by the human brain. Unlike traditional machine learning, these do not have pre-defined characteristics [44]. In spite of the fact that numerous advances have been witnessed in the area of surgery, and mostly measured by the effects of surgical procedures, the key practice of surgery remains a largely low-tech procedure, utilizing instruments and hand tools. Conventional surgery largely relies on the surgeon sensing; touching permits them to differentiate between organs and tissues, which, more often than not, necessitates open surgery. An ongoing revolution is currently in place within surgical technology, with the focus largely being placed on the reduction in the invasiveness of many surgical procedures and on minimizing incisions, the reduction in open surgeries, and the use of flexible cameras and tools during surgery [45]. This minimally invasive surgery is perceived as the future, in spite of the fact that it is still in its early phases. Still, many advancements are yet to be worked upon. In a broader perspective, making surgeries stress-free for patients, reducing the costs incurred and the amount of time. With AI becoming more popular, it is hoped that minimal invasive surgery will be able to focus on contactless methods and provide more tactile feedback. This tactile data processing will usually utilize AI and, more specifically, artificial neural networks so as to heighten the functionality of any available signal translation as well as the interpretation of inherent tactile information [46].

Perceiving the slightest vibrations using piezometric sensors in the vicinity of tissues during surgery can help reduce post-operative risk factors. There is still the possibility that some vibrations cannot be detected due to geometrical and physical constraints. Dynamic physical and chemical characteristics around the target tissue and organ add to the reiteration of predictions. One such idea, implemented by Komeno et al., was that when mechanical vibration is injected into a soft tactile sensor system and then measured by a sensor, the contact state deforms the soft tactile sensors, and the shape or texture of the contacted item should alter vibration propagation. So, the propagated-vibration data should be useful for detecting touched situations. A tiny mounted piezoelectric actuator applies mechanical vibration to a biomimetic (soft and vibration-based) touch sensor as a proof-of-concept. As a test, this method was used to classify sandpaper grit size and slit gap widths and compare their accuracy to sliding motions. Their method classified grit size with 70% accuracy and gap width with 100% accuracy. The results were equivalent or better than conventional sliding comparison approaches [47].

This data analysis, performed using algorithms, is beyond the abilities of human prediction. Additionally, the prediction not only minimizes the risks and surgical course but also the time required. Artificial tactile sensing during liver surgery would offer a couple of advantages in comparison to physical touching and would provide a wider reference library by which to compare standardization and sensation amongst surgeons in relation to continuous improvement, level of training, and quantitative features. The screening of breast cancer presents a classical example where there has been the deployment of artificial tactile sensing. This is seen as being the replacement for clinical breast check-ups and complements medical imaging techniques, such as MRI and X-ray mammography [48].

## 9. Limitations to Implementing Artificial Intelligence

Inherent in the machine learning are logistical issues in implementation, consideration of adoption hurdles, as well as necessary societal or route adjustments. The gold standard for evidence creation is thorough peer-reviewed clinical evaluation inside randomized controlled trials; however, this is not always suitable or possible. Performance measures should be clinically relevant and easily understood by targeted consumers. To prevent patients from being exposed to harmful therapies or denied access to helpful advances, regulation must balance the technological capabilities with the risk of side effects. The use of unbiased, local, and informative test sets is required to enable direct AI system comparisons. Dataset shifts, inadvertent confounder fitting, unexpected discriminating bias, issues of standardization to new populations, and unanticipated negative repercussions of machine learning algorithms on health outcomes must all be considered by algorithm developers [49].

When a new prediction algorithm is introduced, it may create changes in practice, leading to a new distribution. Methods for detecting drift and updating models to improve performance are crucial. Managing this impact requires thorough assessment of performance throughout time to detect difficulties, as well as frequent retraining. Data-driven testing techniques might indicate the most appropriate update strategy, from simple reconfiguration to comprehensive model retraining [50].

Sometimes discriminatory bias is intertwined with generalizability. Machine learning has few blind spots that can reflect social prejudices, resulting in unexpected or undiscovered accuracies in minority groupings, and the potential to reinforce past biases. Studies show that the disadvantages of AI technologies adversely impact populations already disadvantaged by race, gender, and even financial status [51].

## 10. Conclusions

In conclusion, we can state that AI provides medicine and science with enhanced predictions of identification, treatment, and survival following treatment for liver-related diseases as compared to conventional linear models. Innumerous factors determine the surgical course and the care required post-surgery. Major challenges encountered in surgeries are grafting ability, efficiency, and accuracy. ML algorithms have dominantly played a crucial role in developing the prediction models for surgical accuracy. We might need to look deeper using DL in clinical imaging, IR, genomics, and drug discovery. Using DL in analyzing massive biological data can help doctors guide patients and enhance medical wellness in an efficient manner. Issues and obstacles for applying deep learning in computational medicine could be data insufficiency, model interpretability, privacy and ethical issues, and heterogeneity. ML and DL serve as a guideline for future deep learning applications in medicine and health.

It is a universally accomplished fact that improving one’s wellbeing with AI research is difficult. Strong clinical evaluation using surgeon-friendly criteria is required. Identifying algorithmic bias and injustice, reducing lowering risks, promoting generality, and increasing prediction performance of machine learning algorithms, will have revolutionary effects on patients if achieved.

Although AI usage can be especially beneficial in the processing of vast amounts of data, it also aids in the identification of associations and patterns that would not generally be obvious with traditional techniques, especially given the complex nature of biological systems. AI continues to hold a promising role in liver surgery and, more generally, in healthcare research. With an increasing amount of data per patient becoming more accessible, it is crucial understand the extent to which AI can continue guiding clinical decision-making processes. Furthermore, algorithms should be advanced in order to assist the prediction optimization of future outcomes according to the distinctive features of each patient.

## Figures and Tables

**Figure 1 medicina-58-00459-f001:**
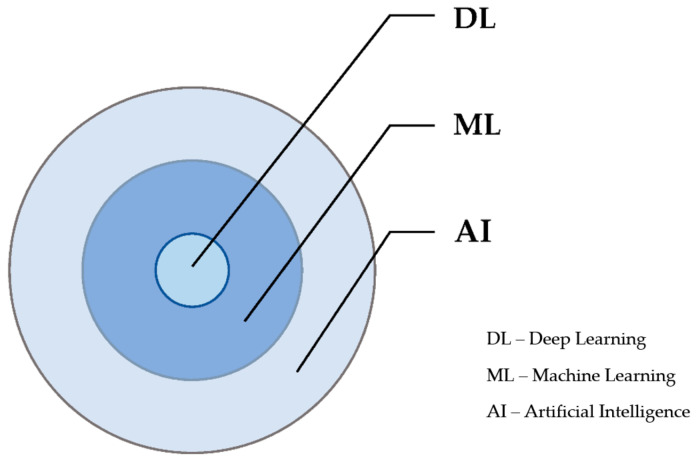
Subfields of artificial intelligence.

**Figure 2 medicina-58-00459-f002:**
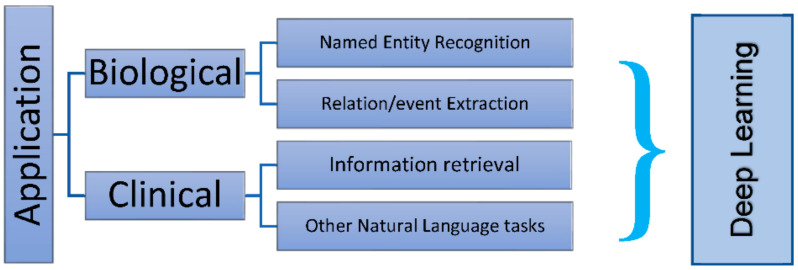
Application of AI in biology.

**Figure 3 medicina-58-00459-f003:**
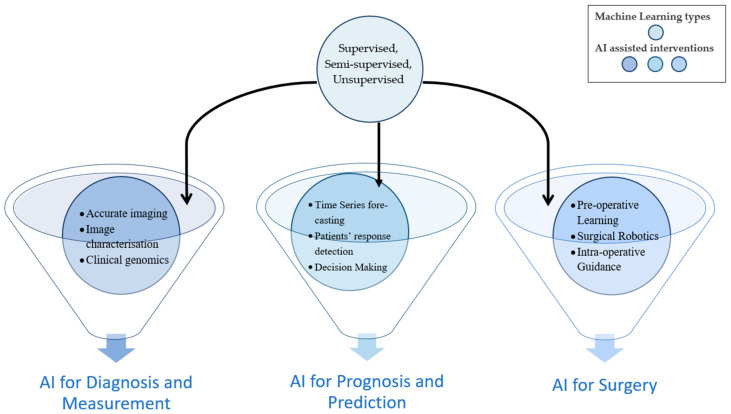
Popular AI techniques for diagnosis, measurement or prediction, prognosis and surgery.

**Table 1 medicina-58-00459-t001:** AI classification using AI image features.


Transform-based	Wavelet-2-H
	Wavelet-2-V
	Wavelet-3-D
	Gabor-1–90
Statistics- based	Volume
	Compactness (compacity)
	Sphericity
	Mean (mean intensity value)
	Mean value of positive pixels/mean positive pixels
	Standard deviation
	Kurtosis (median kurtosis)
	Skewness (median skewness)
	Energy
	Entropy
	Dissimilarity
	Gray-level run-length non-uniformity (gray-level non-uniformity for run)

**Table 2 medicina-58-00459-t002:** HCC radiomics use types.

Objective	Image Type
Diagnosis	CT
Staging and grading	MRI
Therapeutic selection	CT
Prognosis assessment	CT
Prognosis assessment	PET
Surveillance	CT

## Data Availability

The datasets used and/or analyzed during the current study are available from the corresponding author on reasonable request.

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
