# Peer review of "Advancements of Artificial Intelligence in Liver-Associated Diseases and Surgery"

_medicina, 2022, doi:10.3390/medicina58040459_

Round 1

Reviewer 1 Report

The authors summarized the advancements of AI application in liver disease including HCC, CCA, ICC, liver transplantation, and surgeries. However, other important liver diseases like liver fibrosis/cirrhosis, steatosis, or NAFLD/NASH were not discussed.

  1. Full names for AI, ML, or DL in Figure 1 should be given in the legend and on the graph.
  2. please interpret what DNN (line 89), ANN (line 122), SAPS (line 122), AUC (line 123), CNN (line 272), CAD(line 300), was.
  3. As claimed in line 284, three categories (model-, transform-, and statistics-based) were introduced, however, there was no model-based category shown in the table. 1. No detailed explanations were available in the legends to table 1 or 2 for the audience to comprehend what these phrases meant.
  4. there was only figure 3 depicting how AI works for surgery, the working modes for other liver disease diagnosis, prognosis, measurement or prediction were not available.

Author Response

Dear Reviewer 1, thank you very much for pointing out the things you mentioned above. Regarding the points we have done the following :

  • We have changed the Figure 1 as advised
  • For the other important liver diseases – we tried to focus only on carcinoms and diseases in the part of surgery.
  • We have added the self-explanatory full forms of abbreviations as advised by you and gave a short explanation for each term.
  • The model based categories were discussed, but not in the table in the referred article. To avoid misinterpretations, « MODEL based » is removed from our article.
  • And we have finally modified the diagram to include how AI works for other activities like diagnosis and measurement, prognosis and prediction as suggested.
  • We have added liver fibrosis/cirrhosis and NAFLD

Thanks

Reviewer 2 Report

Line 58 - 'inhibits'? Did you mean exhibits

Line 62 - Please revise to 'advent of treatment'

Line 90 - Please expand DNN

Line 122 - Please expand SAPS and ANN

Lines 162-165 - "With each of these factors, there exist both ineffective and effective methods for dealing with them, and which ultimately lead to both positive and negative results, respectively"  - The words positive and negative need to interchanged. 

Lines 266-2267 - In addition to viruses like hepatitis B and C, other causes of chronic liver disease like alcoholic liver disease can also lead to cirrhosis. Please revise this statement. 

Section 7.3 - I would include colorectal liver metastasis (CRLM) under a separate subheading 

Summary: There is a lot of redundancy in the article. Under every subheading, there is mention of advantages of artificial intelligence (AI) rather than pointing out advantages associated with the specific liver condition. For instance, lines 383-386 (this has been discussed several times in the article).  And I would add few lines discussing limitations or drawbacks of AI

Author Response

Dear Reviewer, we thank you for your advices and have done the following in order to get your approval for our work :

  • We corrected and revised the sentences you have mentioned, thank you for pointing out on that.
  • We have added the full forms of abbreviations and gave a short explanation for that.
  • In addition to that we have created a new subheader as suggested by you and have revised our statements.
  • Finally, we have added a dedicated heading along with few references regarding the limitations and drawbacks of AI.

I strongly believe that the paper is now revised best as per the reviewers suggestions after removing the redundant content, changing the diagram, adding a new heading of AI`s limitation and by adding full forms of few abbreviations.

Thanks

Round 2

Reviewer 1 Report

No more comments.

Reviewer 2 Report

I commend the authors for taking note of the recommendations/suggestions by the reviewers and making necessary changes. I would like to congratulate the authors on such a well-written manuscript.